# Image-graph-image auto-encoding:
## *a conceptual study with symbolic shape classification*

### Abstract

This work presents basic research on convolutional neural networks that learn to predict explainable scene graphs from input images without external supervision during training. Unlike existing approaches following a fully-supervised training paradigm, thereby requiring meticulous annotations, we are the first to present a self-supervised approach based on a fully differentiable auto-encoder in which the bottleneck is the graph that corresponds to the input image. To demonstrate the unique conceptual properties of our graph auto-encoder, we apply it to an example task that performs simple rule-based shape classification using only the information in the graph, and we show that our approach allows for the successful classification of shapes that are never seen during training. We report exploratory findings of our research in which the presented approach is applied to elementary line drawings depicting single shapes with limited complexity. We show that our approach exhibits comparable performance to a fully-supervised graph parser baseline, and generalizes significantly better than a conventional image classifier. Although extensive future research is needed to bring our approach to complex natural images, we believe it makes a valuable conceptual step in bridging deep neural networks with graph-based symbolic knowledge representations.

## 1 Introduction

Over the past decade, neural networks and deep learning have significantly advanced the capability of machines in the field of scene understanding: object-wise from image classification Russakovsky et al. (2015) to object detection Ma et al. (2020), and pixel-wise from semantic segmentation Shelhamer et al. (2017) to panoptic-part segmentation de Geus et al. (2021). Most deep learning approaches follow a similar paradigm: 1) for a certain task, a large-scale dataset with the input data and its corresponding targets are prepared; 2) a deep neural network with millions of parameters is designed for the task; 3) the network is trained by feeding the input data and back-propagating the gradients from the designed loss computation. This deep learning paradigm is so powerful that it has become the dominant method in most scene understanding tasks. However, some common concerns still exist, which inhibit further evolution of deep-learning-based artificial intelligence, *i.e.*, the lack of *generalizability* and *explainability*. Currently, neural networks are not able to learn knowledge as efficiently and effectively as humans do. Massive annotations along with multiple regularization methods are required for most of the mid-level or high-level scene understanding tasks, to *partially* overcome the generalizability issue. This issue is less significant when the task remains the same and the domain gap is relatively small, *e.g.*, urban semantic segmentation across multiple environmental conditions. However, it will be challenging if the gap is relatively large, *e.g.*, generalization from urban outdoor Cordts et al. (2016) to indoor scenarios Silberman et al. (2012). Other than generalizability, for most deep learning models, the learned intermediate features are often non-explainable floating-point tensors, which means all information is predominately contained in a 'black box'. This non-explainable property is inhibiting the usage of deep learning in many critical real-life scenarios. Recent research, which is discussed in Section 2, has been carried out to tackle the aforementioned challenges, yet deep learning is still far from inherent 'human-like' generalizability and explainability for neural networks.

To take one step forward towards overcoming the aforementioned limitations, we set the basic research goal to learn generalizable and explainable graph representations from the input image data, preferably without

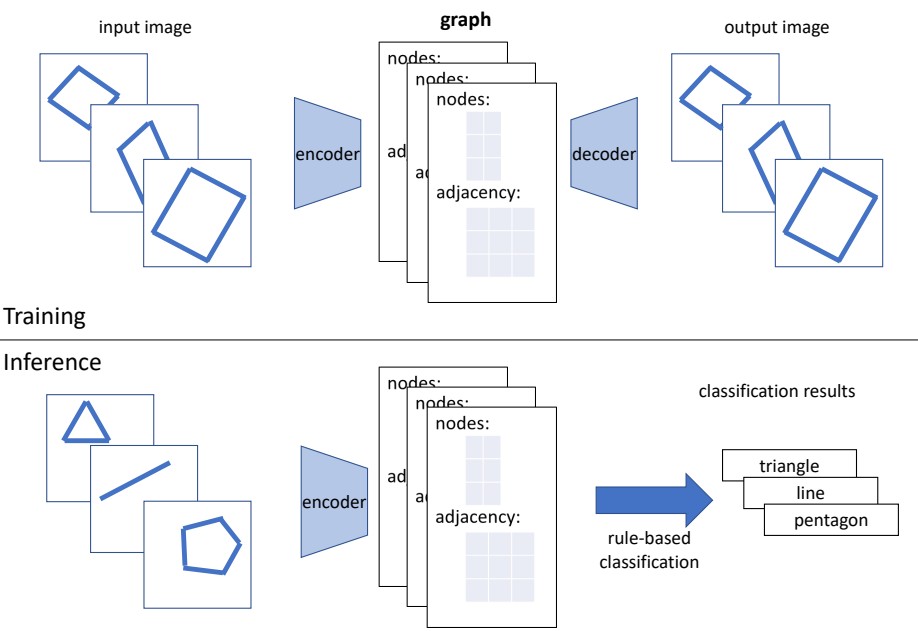

Figure 1: Overview of the proposed approach. During training (upper part), the auto-encoder is able to learn to extract a graph encoded by a node and adjacency matrix from an input image, using only image-level self-reconstruction loss. During inference (bottom part), the learned encoder is able to generalize to unseen shapes without any adaption process, and a rule-based classifier is able to predict the shape label using the predicted graph and prior knowledge of the unseen shapes. The encoder and classifier compose an explainable and generalizable hybrid neural-symbolic system.

massive manual annotations. This is a complex challenge and consequently, we take a step back with respect to possible applications and focus on translating line-drawings of simple basic shapes into graphs. Inspired by the polygonal mesh techniques Botsch et al. (2007) in computer graphics (CG), where a complicated scene is always created by simple polygon elements, *e.g.* triangles, we consider the scene understanding task as the reverse process of CG, where the input scene could be abstracted and parsed as a combination of simple elements. With this assumption, in this work we try to parse the input image containing simple shapes into a set of pre-defined visual elements, *i.e.*, connected line segments. In this case, line segments are basic building blocks of the image, allowing the image to be parsed, even if the high-level scene is changed and is never seen during training. Thus, this approach allows the scene parsing to be more *generalizable* with respect to the scene content it can handle. Furthermore, we efficiently model the line segments and their connectivity as a scene graph, allowing the representation to be human-interpretable, see the upper part of Figure 1. In this work, the scene graph and the set of interconnected line segments are interchangeable.

We achieve to predict the scene graph representation by learning to auto-encode the images, which is detailed in Section 3, thus not requiring manual annotations. At its core, it learns to translate image content into a scene graph containing a set of nodes and an adjacency matrix. The entire learning is self-supervised using a fully differentiable auto-encoder in which the encoder consists of several neural network modules and the decoder uses techniques from spatial transformer networks Jaderberg et al. (2015). The entire network is trained end-to-end by minimizing the pixel-wise similarity loss between the input image and the image generated by the decoder. During inference, only the learned encoder is used for graph prediction. Other than the raw training images, the only needed information is the template set of the visual elements (line segments in our case) for the decoder and no further annotation is required, even when testing the trained model on novel unseen shapes.

The extracted scene graph from the encoder is the description of the input image at a highly abstract level. It can be processed further for high-level scene understanding tasks, and can serve as the bridge between neural networks and symbolic systems. We also showcase this with a simple rule-based classifier that takes a

scene graph as input and predicts the shape label, see the bottom part of Figure 1. As long as the classifier has the basic symbolic knowledge of the shape, for example "a triangle has three nodes that are connected with lines", the classifier, similar to humans, is able to provide the corresponding shape label for the input scene graph without having seen this particular shape before. This is achieved because the learned scene graph is fully *explainable* such that we humans are able to describe and encode the knowledge in the classifier. This classifier, together with the previously mentioned encoder, composes an explainable and generalizable hybrid neural-symbolic system, which will be formalized in Section 3, with several experiments showing its behavior in Section 5.

To research and demonstrate our approach, we use a synthetic dataset that contains several line drawings of simple basic shapes. The details of the dataset and its corresponding experimental results are presented in Section 4 and Section 5. While the current capability of our approach is limited to simple line drawings, each containing only a single basic shape, we believe it is an important conceptual step in developing more generalizable and explainable deep scene parsing methods that scale over many different scenarios. Although extending our approach to more complex natural images will inevitably require a level of manual annotation, in this work, we focus our research strictly on the self-supervised auto-encoder paradigm.

In summary, the contributions of our presented basic research are:

- We propose, to the best of our knowledge, the first neural network that can learn to translate images into scene graphs by means of an image-graph-image auto-encoder;

- The translation into a scene graph, *i.e.*, the bottleneck representation of the auto-encoder, is achieved via a novel network design, which does not require manual annotation for training;

- We explore the advantages of our self-learned scene graph representation, *i.e.*, its generalizability and explainability, by applying it to a classification task for unseen shapes.

## 2 Related work

**Learning generalizability and explainability** Neural networks tend to overfit the training data and lack in generalizability to unseen data Zhang et al. (2017): a performance drop can be easily observed when the input data has a minor domain gap compared to the training data. Domain adaption research has been carried out to improve the network's generalization ability on multiple datasets within the same setting for the same task, *e.g.*, semantic segmentation for autonomous driving in different weather conditions. Meta-learning, including few-shot Xu et al. (2021) and one-shot Kasnesis et al. (2021) learning, is proposed to push the generalizability of networks even further: the network is expected to work even if the task settings are significantly changed, *e.g.*, to classify a new unseen label. This is achieved by an adaptation process with limited exposure to the new task configuration.

Instead of approaching the generalizability under the conventional deep learning paradigm, *i.e.*, learning black-box feature tensors, representation learning at a symbolic level might naturally exhibit better generalizability. This is because symbolic representations, *e.g.*, scene graphs, are fully explainable, thus non-relevant domain-specific information can be identified and neglected if necessary. In this work, we showcase this advantage by performing shape classification for unseen shapes, which only utilizes a simple shape description in the form of rules, *e.g.*, "a triangle has three nodes that are all connected by lines". This is achieved without an additional mini-learning session, and as such, it can be considered as a particular rudimentary form of zero-shot learning Wang et al. (2019).

**Neural-symbolic computing** The lack of explainability and generalizability prohibits the wider deployments of deep learning-based AI to critical tasks that involve safety and ethics. One motivation of neural-symbolic AI Garcez & Lamb (2020) is to solve the lack in explainability and generalizability of neural networks with the help of symbolic reasoning. Recent work is mainly focusing on transforming symbolic representations into the format of tensors, such that neural networks are able to perform symbolic reasoning with canonical tensor computations. For instance, a neural tensor network (NTN) Socher et al. (2013) is proposed to reason over relationships between two entities using relation knowledge embedding. Other than

relational knowledge, logic tensor networks (LTN) are able to encode logical formulas into neural networks Serafini & Garcez (2016).

An alternative direction is to have a hybrid neural-symbolic system in which a neural network and a separate symbolic system can interact with each other, as long as a proper interface can be designed, as proposed in Garcez & Lamb (2020), and demonstrated by various specific tasks, such as video action reasoning Zhuo et al. (2019) and visual Q&A Shi et al. (2019). In this work, we follow a similar direction and propose our graph auto-encoder which encodes image information into a graph that subsequently acts as the neural-symbolic interface. This graph is fully explainable and thus can be processed by any symbolic system, which is showcased with a simple rule-based classifier in our experiments of Section 5.2.

**Self-supervised learning via auto-encoding** To reduce the dependency on expensive manual annotations, unsupervised or self-supervised approaches have been widely investigated. Most methods follow an auto-encoding fashion: the encoder takes the input data for representation learning at the bottleneck, and the decoder is designed to reconstruct the input information from the bottleneck's representation. Thus, the only main supervision is the self-reconstruction loss and no annotation is required. The learned representations can include individual objects and their information, physical factors, *etc.*. In Burgess et al. (2019) a variational auto-encoder is trained together with a recurrent attention network decomposing the objects in an image. More recently, the work in Yang et al. (2020) went one step further to decompose the image into objects and enable object manipulation without requiring object-level annotations. Also, in Wu et al. (2020) the proposed encoder-decoder framework factors each input image into depth, albedo, viewpoint, and illumination, using a single unsupervised reconstruction loss.

The mentioned approaches achieve unsupervised learning of scene representation mainly by carefully designing the auto-encoding framework. Our approach has the same philosophy, but is more aggressive in the representation-shifting that is performed: the self-learned representation (scene graph) is further away from the input representation (image) in terms of the data abstraction and the level of scene understanding, while in other works Burgess et al. (2019); Yang et al. (2020); Wu et al. (2020) the self-learned representations are at a lower level of abstraction and thereby closer to image-like representations. Although our approach is still limited to scenes containing a single simple basic shape, it is, to the best of our knowledge, the first neural network-based auto-encoder that learns to encode image information into a scene graph using only self-supervision.

## 3 Methodology

As illustrated in Figure 1, the proposed approach is composed of two phases: *learning* and *inference*. The auto-encoder first learns the graph representation from images without any external supervision. During the inference, an additional knowledge-based shape classifier is introduced, in combination with the learned encoder, to form a hybrid neural-symbolic classification system. In this section we first provide a formal definition of both learning and inference phases in Section 3.1, and then detail the novel design of the proposed auto-encoder and its training settings in Section 3.2 and Section 3.3, respectively.

To present the idea, we take a simple toy task with a synthetic dataset as an example. The dataset contains a few visible shapes constructed by undirected graphs. The nodes in the graphs are the end-points of the line segments and the edges are defined as the binary connectivity between two nodes, *i.e.*, the presence of a line. Please see some visualized samples in Figure 5 to better understand the task.

### 3.1 Problem definition

### 3.1.1 Learning scene graphs via auto-encoding

We start with the conventional problem definition of the scene graph generation task Johnson et al. (2015) and then discuss the similarities and differences in our setting.

Given an input image $I$, we assume that there exists a scene graph $G = (V, E)$ with $V$ being a set of nodes corresponding to localized object regions (in our task, line end points) in image $I$, and $E$ being the edges

representing the relationships between two object regions $V$ (in our task, a connected line segment between two end points). Note that each element $v_i$ and $e_{ij}$ in $V$ and $E$, where subscripts $i, j$ denote node indices, could have one of multiple semantic labels. Thus, the scene graph generation can be formulated as the mapping $f : I \mapsto G$. Most approaches Xu et al. (2017); Li et al. (2018); Qi et al. (2019) factorize it mainly into two sequential sub-mappings, $i.e.$, $f = f_E \cdot f_V$, where

$$
\begin{aligned}
f_V &: I \mapsto V \\
f_E &: (I, V) \mapsto E.
\end{aligned}
\tag{1}
$$

$f_V : I \mapsto V$ maps the image to a list of nodes and is often accomplished by object detection frameworks, and $f_E : (I, V) \mapsto E$ estimates relationships between nodes using image information, $i.e.$, it is a relationship classifier. Both of these mapping processes are usually performed by neural networks. In the conventional fully-supervised setting, for each training sample the ground truth nodes $\bar{V}$ and edges $\bar{E}$ are provided such that the neural networks performing the mapping $f : I \mapsto G$ can be trained using the ground truth.

If the ground truth $\bar{V}$ and $\bar{E}$ are not available, the conventional supervised approaches cannot be applied, which holds for most tasks and datasets. On the contrary, we opt to extend the task of scene graph generation to image-graph-image auto-encoding. The encoder $f : I \mapsto G$ is expected to perform the same task as in scene graph generation Johnson et al. (2015). Only in our case, the supervision is not provided. The decoder $g : G \mapsto I$ takes the intermediate graph information as input, and re-generates the input image in a fully differentiable manner. By placing the fully differentiable encoder and decoder behind each other, techniques from auto-encoding can be used to learn the graph information in a self-supervised manner.

Formally, our goal is to have two mapping functions $f$ and $g$,

$$
\begin{aligned}
\tilde{G} &= f(I) = f_E \cdot f_V(I) \\
\tilde{I} &= g(\tilde{G}),
\end{aligned}
\tag{2}
$$

such that

$$
f, g = \arg\min_{f,g} \mathcal{L}(I, g(f(I))),
\tag{3}
$$

where $\mathcal{L}(\cdot, \cdot)$ is a similarity measure of two images.

### 3.1.2 Inference with hybrid neural-symbolic system

During inference, the decoder $g : G \mapsto I$ is not needed, and the encoder $f : I \mapsto G$ continues to extract scene graphs $\tilde{G}$ from novel unseen images. One could make use of this explainable graph representation for multiple different applications. To demonstrate the neural-symbolic interface capability of the obtained graph, as an example, in our experiments we propose a simple knowledge-based symbolic shape classifier $h : G \mapsto L$, that takes the predicted scene graph $\tilde{G}$ as input, and predicts high-level shape labels. Together, they formulate a hybrid neural-symbolic system with the scene graph being the interface. Formally the system can be written as

$$
\begin{aligned}
\tilde{G} &= f(I) = f_E \cdot f_V(I) \\
\tilde{L} &= h(\tilde{G}),
\end{aligned}
\tag{4}
$$

where $\tilde{G}$ and $\tilde{L}$ are the predicted scene graphs and shape labels.

The knowledge-based symbolic shape classifier $h : G \mapsto L$ simply stores the knowledge of different shapes as adjacency patterns, and predicts the shape of each input graph by adjacency pattern matching. Specifically, for each input graph $\tilde{G}$, the shape classifier first examines the number of nodes in $\tilde{G}$. Based on the number of nodes, the corresponding adjacency matrix is further examined by several graph isomorphism tests with the adjacency patterns of known shapes in the knowledge base. Once a certain graph is matched with a certain shape pattern, the corresponding label from the knowledge base is retrieved. For example, if the graph has three nodes in total, and in the adjacency matrix the nodes are all connected with each other, then the predicted class label retrieved in the knowledge base for this node/adjacency pattern is 'triangle'. If none of the node/adjacency patterns is matched, the classifier will predict 'unknown' class, since it is beyond the knowledge encoded in the knowledge base.

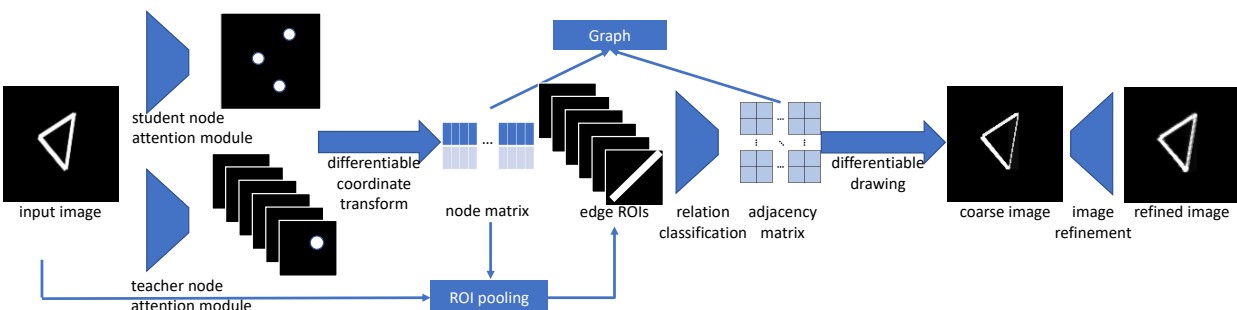

Figure 2: The framework of the proposed auto-encoder. The entire framework is trained end-to-end and using self-reconstruction supervision. Please see Section 3.2 for more details.

Please note that here we use a very simple symbolic classifier to showcase that the learned scene graph representation can serve as a bridge between neural networks and conventional symbolic systems. There is no doubt that more advanced symbolic systems, which are out of the scope of this research, can be applied to further boost the performance and can provide more functionalities.

## 3.2   Network

The overall design of the network for the image-graph-image auto-encoding task is visualized in Figure 2. It mainly consists of five differentiable modules, *i.e.*, *node attention*, *coordinates transformation*, *relation classification*, *differentiable image synthesis*, and *image refinement*. The five modules are sequentially connected and trained in an end-to-end manner using image self-reconstruction loss.

### 3.2.1   Encoder: from image to graph

As discussed previously, the encoder part of the network predicts the scene graph from the input image, which is formalized as the process $f = f_E \cdot f_V : I \mapsto G$. The encoder is composed of three sequential parts, namely *node attention*, *coordinates transformation*, and *relation classification*, and all of these modules are end-to-end differentiable. The output of the encoder, *i.e.*, the bottleneck of the auto-encoder, is the graph represented by two matrices, *i.e.*, the node position matrix and the adjacency matrix.

**Node attention**   At the early stage, two node attention modules are implemented in parallel with several convolutional layers from ResNet-50 He et al. (2016): one for self-learned node attentions (teacher attention) and the other for learning the attentions from the self-learned teacher (student attention), see left upper part in Figure 2. The functionality of the node attention modules is to take the image as input and produce heat maps of expected node locations. Here we discuss the high-level design, and for the detailed implementation of two node attention modules, please see the released code authors (2021).

For the self-learned teacher attention module, we take features from the second residual block in ResNet-50 and reduce the output channels of two blocks to 128 and 64, respectively. On top of the residual blocks, we have two extra convolutional layers to predict the node attention maps $M_{att}$, with their spatial size reduced by a factor of 4 due to the previous pooling operations, and with their number of channels being the *pre-defined* maximum number of nodes $N_{max}$. Each channel in node attention maps $M_{att}$ is passed through a 2-D softmax layer, such that the summation of all the elements strictly equals to 1, which is essential for the later computation of node coordinates. Note that here we expect that the attention channels are able to provide channel-wise heat maps indicating the detected node locations. However, no supervision or extra information is supplied to the teacher attention module, and the network is able to learn by itself from the final reconstruction loss by means of gradient back-propagation.

Although the teacher attention module is able to learn where to attend the existing nodes without supervision, some limitations exist: 1) the pre-defined maximum number of nodes $N_{max}$ will create redundancy and

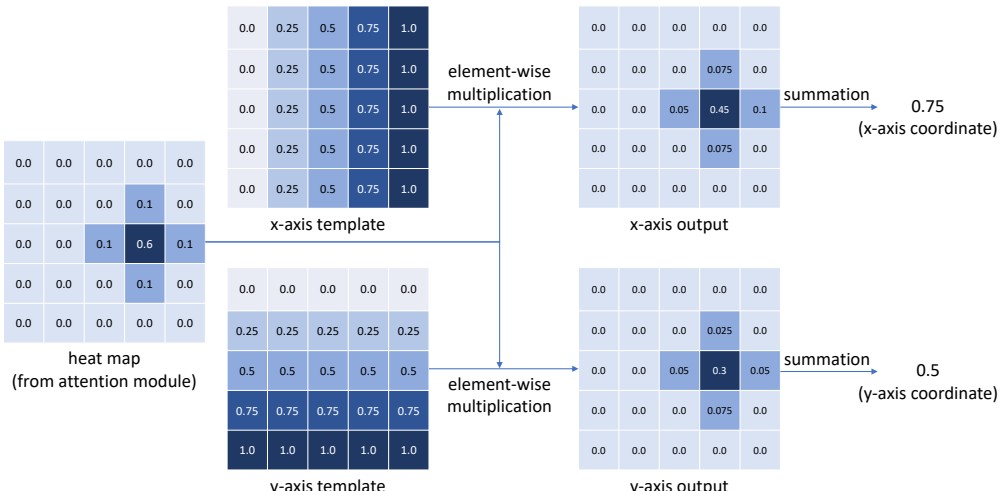

Figure 3: The process of coordinates transformation using *differentiable spatial to numerical transform* (DSNT) Nibali et al. (2018). Please see Section 3.2 for more details.

ambiguity when the real nodes are less than the maximum; 2) it cannot generalize to scenarios with nodes more than $N_{max}$. Thus, we propose the use of a student attention module that learns from the teacher but in a more generalizable fashion. The student module has a similar network structure, the only key difference is that the attention map has a single channel and the training target of this channel is the sum over the attention channels of the teacher attention module. Then, a yolo-like Redmon et al. (2016) grid proposal and non-maximum suppressing Neubeck & Van Gool (2006) are used to extract the node matrix in the same format as the teacher module. Since the student attention is based on region proposals, it does not rely on the pre-defined maximum number of nodes $N_{max}$, and naturally generalizes to various numbers of nodes. This process is not fully differentiable thus it cannot learn directly from the image reconstruction loss like its teacher. Instead, the student attention module is learned simultaneously from the teacher's prediction with simple binary cross entropy loss. Thus the teacher-student approach allows for 1) self-learning via the teacher and 2) generalization via the student. During inference, the student attention can be used in replacement of the teacher seamlessly, and it exhibits even better performance in some scenarios, which will be showcased in experiments in Section 5.2.

**Coordinates transformation**  The functionality of this module is to transform the local maxima in the heat map to node coordinates for the graph representation in a fully differentiable manner. Being different from common detection frameworks, in our setting the coordinates of nodes are not provided as the ground truth. Thus all the modules must be differentiable such that the earlier teacher node attention module can learn via back-propagation. It is non-trivial to differentiably transform the positional information in the heat maps to numerical coordinates and here we use *differentiable spatial to numerical transform* (DSNT) Nibali et al. (2018) to perform the transformation. Due to the usage of the previously mentioned 2-D softmax layer, for each channel, the heat map can be seen as a 2-D probabilistic distribution of a certain node. Thus, one can create two fixed template maps containing the numerical coordinates for each pixel, in horizontal and vertical directions, respectively. With element-wise multiplication and summation, the numerical coordinates for each node can be computed, please see Figure 3 for an illustration. At this stage, there exist no trainable parameters, and the computation, which is detailed in Nibali et al. (2018), is fully deterministic and differentiable.

**Relation classification**  Once the coordinates of the nodes are computed, the relation classification is performed on all possible pairs of nodes, which results in an adjacency matrix as part of the predicted graph. To implement this, a set of regions-of-interest (ROIs) for classification is constructed by combining two node coordinates into a bounding box. Then ROI pooling Ren et al. (2017) is applied on the input image, which

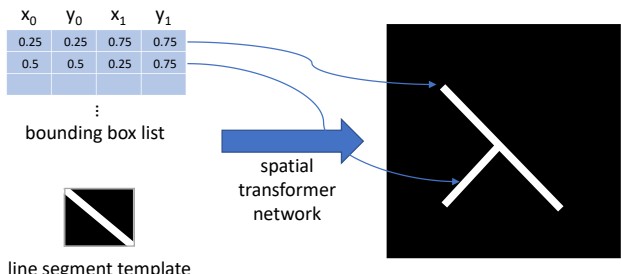

Figure 4: The structure of the differentiable drawing module.

creates a batch of local image patches of size $16 \times 16$ for further edge prediction by a relation classifier. This classifier consists of two convolution layers (kernel size 3 and output channels 32 and 16, respectively) with Batch Normalization, ReLU activation, and max-pooling. The output features of these convolutional layers are flattened and processed by two fully-connected layers for final relation classification. The supervision is provided via image reconstruction loss, which is detailed in Section 3.3, since the auto-encoding pipeline is fully differentiable.

Please note that in our toy task the relation represents the binary connectivity, while it can be extended with semantic information with little extra effort. The predicted nodes' connectivities, together with the previously computed nodes, make up the output of the encoder, *i.e.*, a graph encoded by a node position matrix and a node adjacency matrix.

### 3.2.2 Decoder: from graph to image

**Differentiable image synthesis** The task of the decoder $g : G \mapsto I$ is to reconstruct the input image given the graph information provided by the encoder. The key is to ensure that the reconstruction flow is fully differentiable such that the whole pipeline can be trained together end-to-end. Inspired by Johnson et al. (2018), we consider a template-based image synthesis approach, which is referred to as *differentiable drawing*. Its design is presented in Figure 4.

We create a template set that contains all the pre-defined relationships that could exist in the dataset, and in our task, the template set contains only a line segment, See Figure 4. For each forward-pass of the network, the differentiable drawing module walks through all the edges in the predicted adjacency matrix, and for the edges that need to be visualized in the image, a spatial transformer network Jaderberg et al. (2015) is implemented to copy and draw the corresponding edge template onto the canvas, with differentiable bi-linear interpolation. The differentiable drawing module is able to create a coarse reconstructed image $\tilde{I}_{\text{coarse}}$ on a blank canvas. This process can be written as $g_{\text{coarse}} : G \mapsto \tilde{I}_{\text{coarse}}$.

**Image refinement** On top of the coarse image $\tilde{I}_{\text{coarse}}$, a refinement network $g_{\text{refine}} : \tilde{I}_{\text{coarse}} \mapsto \tilde{I}_{\text{refine}}$ is applied for further post-processing, which does not change the structural information but modifies and eliminates the textures and styles difference between the created coarse image and the real input image. In our implementation, the refinement network contains three convolution layers with kernel size 3, intermediate channel size 16, and PReLU activations between every two layers.

### 3.3 Training

The entire framework is trained end-to-end without supervisions that require manual annotations. To achieve this goal, we apply three losses during training: 1) the main image reconstruction loss on the refined image, 2) the auxiliary reconstruction loss on the coarse image, and 3) supervision loss for the student attention module so that it can learn from the self-learned teacher.

**Main image reconstruction loss** As the main image reconstruction loss, a structural similarity index measure (SSIM) Wang et al. (2004) is used, which encourages the decoder to reconstruct the input image and

thus encourages the encoder to understand the image in terms of graphs in an implicit manner. At the core, SSIM compares the perceived change in structural information, including luminance and contrast, which exhibits behavior closer to the human visual system and is fully differentiable. For detailed interpretation and implementation, please see Wang et al. (2004). We apply SSIM loss with a multi-scale (MS-SSIM) setting on the refined images in the experiments, which is formulated as

$$\mathcal{L}_{\text{main}} = \text{MS-SSIM}(\tilde{I}_{\text{refine}}, I) \tag{5}$$

where $\tilde{I}_{\text{refine}}$ and $I$ are the reconstructed refined image and input image, respectively.

**Auxiliary image reconstruction loss**   To further improve the reconstruction quality, in addition to the MS-SSIM loss on the refined image, we apply the same loss on the coarse image. It is intended to stabilize the adjacency prediction and regularize the functionality of the refinement network, which will be verified and discussed in Section 5.3.2. Similarly, the auxiliary loss can be written as

$$\mathcal{L}_{\text{aux}} = \text{MS-SSIM}(\tilde{I}_{\text{coarse}}, I) \tag{6}$$

where $\tilde{I}_{\text{coarse}}$ and $I$ are the reconstructed coarse image and input image.

**Student node attention loss**   The conventional binary cross entropy (BCE) loss is implemented for student attention module learning. The learning target $M_{\text{stu-target}}$ is dynamically generated from the prediction of the teacher module $M_{att}$ by normalizing the maximum value for each channel and then stacking channels together as a single one:

$$M_{\text{stu-target}} = \sum_{i}^{N_{\text{max}}} M_{att}[i]/\max(M_{att}[i]) \tag{7}$$

Then the target map $M_{\text{stu-target}}$ is binarized and the loss $\mathcal{L}_{\text{student}}$ for training the student node attention modules can be formalized as

$$\mathcal{L}_{\text{student-att}} = \text{BCE}(\tilde{M}_{\text{student}}, M_{\text{stu-target}}) \tag{8}$$

**Combined loss**   Formally, the overall loss can be written as

$$\mathcal{L} = \mathcal{L}_{\text{main}} + \lambda_1 \cdot \mathcal{L}_{\text{aux}} + \lambda_2 \cdot \mathcal{L}_{\text{student-att}} \tag{9}$$

where $\lambda_1$ and $\lambda_2$ are the weights for loss balancing and is empirically set to 0.1 and 1 in our experiments. The ablation study for the losses is presented in Section 5.3.2.

## 4   Experiments

We perform the following experiments to demonstrate and verify the proposed method: 1) we compare the results of our unsupervised approach to a supervised *graph parser* baseline and a *CNN classifier* baseline, both qualitatively and quantitatively; 2) we directly apply the trained models on an unseen dataset with multiple unseen shapes, and verify the generalizability of our approaches; 3) we perform several ablation studies of our approach, including varying the maximum number of nodes and different loss settings.

In this section, we detail the experimental settings including the datasets, baselines, metrics, and the training methodologies, and in Section 5, we present the results and their analyses.

### 4.1   Datasets

We create two synthetic datasets, *Simple Shape Seen* dataset and *Simple Shape Unseen*, to research and demonstrate the proposed auto-encoding method. The *Simple Shape Seen* dataset contains 50k images that only contain rectangles. Each shape is processed with random affine transformation (including scale, rotation, shear, and translation) and is drawn on an empty black canvas with the size being $128 \times 128$ pixels.

The dataset is split into three parts in the experiments, *i.e.*, train (45k), val (1.5k), and test (3.5k). In contrast, the *Simple Shape Unseen* dataset has 3.5k images, containing several shapes with similar random affine transformations that are not presented in the seen dataset. The unseen shape includes *line*, *triangle*, and *pentagon*. This dataset is not used for training the network, instead it is for testing the generalizability of the learned network. Please see Fig 5 and Fig 7 for some visualized samples. Note that for the Simple Shape Seen dataset, each sample has a fixed number of nodes, *i.e.*, 4 (rectangle), yet for samples in the Simple Shape Unseen dataset, the number of nodes varies, including 2 (line), 3 (triangle), and 5 (pentagon).

## 4.2 Baselines

We propose two baselines in the experiments, namely *graph parser* and *CNN classifier*, to validate the proposed approach.

**Graph parser**   We first provide a *fully-supervised* baseline that has the same functionality as our approach, *i.e.*, parsing input images into graphs. It shares a similar design with our encoder network, but has access to the ground truth graph during training. To have a fair comparison, the capacity of the baseline network, is aligned with our proposed network as much as possible, including the overall structure, number of layers, and their channels. Since the ground truth graph is provided, the decoder is no longer needed for the baseline, and we directly apply two supervisions to the encoder's outputs, *i.e.*, the node matrix and the adjacency matrix. Furthermore, since the teacher attention module is no longer needed to provide the self-learned target attention maps, we only use the student attention module in the graph parser baseline.

To train this baseline, first, the supervision for the node attention channels is provided by creating a ground truth heat map with Gaussian kernels at the node ground truth locations, for computing the binary cross entropy loss. Second, for each step, we perform the node matching between the prediction and the ground truth, to re-generate the temporary ground truth adjacency matrix aligned with the correct temporary nodes' order. This allows for a cross entropy classification loss to be computed. With the aforementioned two supervisions, *i.e.*, pixel-wise binary cross entropy node attention loss and binary cross entropy classification loss, the graph can be learned in a fully-supervised manner.

Please note our goal is not to outperform this fully-supervised counterpart using our self-supervised approach. The goal is to obtain a reasonable upper bound that our self-supervised graph-based approaches can be compared with.

**CNN classifier**   To further validate the explainability and generalizability of our graph-based approaches, we also include a conventional *CNN classifier* as the second baseline. In this case, graphs are not relevant as it is only trained to directly predict the shape labels of input images in a conventional fully-supervised manner. Like the previous baseline, the capacity of the network is also aligned with our approach as much as possible. This baseline is motivated from the perspective of shape classification tasks. Unlike the graph-based neural-symbolic systems, the CNN classifier cannot generalize to unseen shapes that are not exposed during training, which is experimentally demonstrated in Section 5.2.

## 4.3 Metrics

We use two metrics for evaluating the performance: 1) *triplet matching score* and 2) *shape classification accuracy*. The first one decomposes a scene graph into triplets and reflects the detection quality for each triplet in graphs; the second one reflects the higher-level quality of graph prediction, which is the accuracy of shape classification. As the CNN classifier does not provide graphs as output and predicts the shape label directly, we only report its shape classification accuracy in the experiments.

*Triplet matching score*: To evaluate the quality of scene graph, the image-wise SGGen metric is often used in related work Xu et al. (2017); Li et al. (2018); Qi et al. (2019). The idea is to organize the prediction and ground truth graphs as two sets of triplets with the format of <*object, relationship, object*>. Then triplet matching is performed to compute the conventional Recall@k metric. The main reason only the recall is used and not also precision in previous work is that the manual annotations are sparse and it is not possible to annotate all existing relations between objects in images. However in our dataset, since the images are

created from a given graph, we are able to capture the complete graph information. Thus, in our work, the precision metric is also reported. In our evaluation protocol, like previous work Xu et al. (2017); Li et al. (2018); Qi et al. (2019), we also consider the task as detection of triplets, and use F1-score with underlying metrics being precision and recall. A true positive sample is found if all three elements match with one of the ground truth triplets.

Since our model might have multiple predictions for the same triplet (when using teacher attention modules and the maximum number of nodes is redundant), we process the raw prediction and delete the redundant triplets before evaluation. The recall is computed without sorting the confidence and all the predicted triplets are used for the computation of precision. In this sense, our F1-score metric containing precision and recall covers the previously used SGGen, which is essentially recall, and is more challenging in terms of triplet redundancy evaluation, which is essentially the precision.

*Shape classification accuracy*: The previous metric only evaluates the raw graph predictions in terms of triplet matching, which remains at the local level and only focuses on the performance of the auto-encoder. For the shape classification accuracy, the performance is evaluated directly on the predictions by the combined neural-symbolic system or CNN classifier baseline. Thus, this metric reflects the overall performance of the complete system during the inference phase. In practice, we collect the label predictions from different approaches, and compare them with the ground truth labels for accuracy computation.

This metric is used along with the conventional triplet matching metric, and is more challenging for graph-based approaches to some extent: if one triplet is mispredicted, the shape will be absolutely wrong while the F1-score of triplet matching will only degrade slightly.

### 4.4 Implementation details

We train our approach and the baseline using PyTorch, and we use Adam optimizer with batch size $= 32$, $\beta_1 = 0.6$, $\beta_2 = 0.9$, and weight decay $= 0.0001$ for the training of both models. We also train the network in each setting with different random seeds for 5 times, and report the average for each metric. For the *proposed* approach, we train with the initial learning rate 0.0001 for 8 epochs. As for the *graph parser* baseline, we notice that providing the full ground truth will significantly simplify the task, which is expected. Thus, we train the baseline model with the initial learning rate 0.0001 for 5 epochs. Also, the supervision quality for the adjacency matrix is dependent on the quality of the predicted nodes, which is used for generating the temporary ground truth on the fly. Thus for the graph parser baseline, in the first two epochs we train the node attention module only and later we add the adjacency classification loss together with the node attention loss. For the *CNN classifier* baseline, we simply train it for 2 epochs by feeding the ground truth shape labels, given the simplicity of the task. For other implementation details, please see the code in authors (2021).

## 5 Results

### 5.1 Performance on seen data

We first compare the performances (see Table 1) between our self-supervised approach and baselines, when they are trained and tested on the Simple Shape Seen dataset. For graph-based approaches we also present graph visualizations in Figure 5. For our approach, we present the graph prediction performance when using two different node attention sources: the self-learned teacher attention module (tea. att.) and the student attention nodule (stu. att.).

From Table 1, it can be seen that the CNN classifier achieves 100% accuracy, as the test samples are all rectangles, which is identical to the training data. This perfect performance is expected, but comes with a specific limitation: it cannot generalize to unseen shapes unless one re-trains the network with such samples, unlike our graph-based approaches, which is demonstrated in Section 5.2.

As for graph-based methods, our approach *with teacher attention module* exhibits comparable performance to the fully-supervised graph parser baseline: in terms of F1-score, the baseline outperforms our approach by

Table 1: Quantitative results of our approach and the supervised baselines evaluated on *Simple Shape Seen* test dataset. Two baselines require different supervisions during training.

| Method | Sup. | Triplet matching | | | Classification |
| --- | --- | --- | --- | --- | --- |
| | | Precision | Recall | F1-score | Accuracy |
| Ours (tea.) | None | 97.7±0.7 | **98.5**±0.7 | 98.1±0.7 | **96.2**±0.6 |
| Ours (stu.) | None | 94.0±0.6 | 92.5±1.1 | 93.3±0.6 | 66.7±3.5 |
| Graph parser | Graph | **98.5**±0.6 | 98.7±0.2 | **98.6**±0.4 | 91.1±1.7 |
| CNN classifier | Label | - | - | - | 100.0±0.0 |

0.3%, while in terms of shape classification accuracy, ours is better by a margin of 5.0%. This relatively small gap shows the strength of our self-supervised approach: even without any external explicit ground truth, the self-reconstruction is sufficient to provide supervision for the task of graph prediction. Please note that we do not claim that our approach is absolutely better than the graph parser baseline in terms of the performance. One can optimize its design in many aspects and achieve performance improvements. However, we opt to keep the network design the same, to gain more meaningful insights into our self-supervised approach.

For our approach, we also compare the difference between the student and teacher node attention modules. The self-learned teacher attention module exhibits better performance compared to the student counterpart by 4.9% triplet F1-score and 28% classification accuracy. This follows our intuition, since the student module takes the teacher module's prediction as the ground truth, and the acceptable performance drop shows that the training of the student module is effective. Although the self-learned teacher attention module has better quantified performance, the implementation of student attention module is necessary for dealing with unseen shapes, as will be validated in the experiment on the Unseen Shape Dataset (see Section 5.2).

Other than the quantified performance, it is worth emphasizing that unlike many other auto-encoding approaches that process the data in the same format, *e.g.*, image-feature-image translation, in our case, the information in the bottleneck is a conventional graph with nodes in random order with their corresponding adjacency matrix. By creating differentiable transformation modules, as introduced in Section 3, an image-graph-image auto-encoding framework is able to regress the graph information in this canonical format automatically. This property is novel even *without* considering the quantified performance. However, it must be clarified that the unsupervised approach contains several limitations that the supervised counterpart does not share. First, even though we use a simple dataset, it is less efficient for the network to learn under the self-supervised setting, which can be noticed by the number of training epochs used by the two approaches. Second, we notice that the baseline is less sensitive to random effects during training, while our approach, although very rarely, has chances to collapse. The cases in which our approach collapses are considered as outliers and are not included in the quantitative results. Third, if the task is more complicated and challenging, the self-supervised learning would be less effective or potentially even fail, compared to the fully-supervised approach. Nevertheless, our basic research demonstrates that it is possible to learn the graph representation without any manual annotation using auto-encoders, which opens up new avenues for future research.

## 5.2 Inference on unseen data

As discussed in Section 1 and 2, generalizability and explainability are the key limitations for contemporary (fully-supervised) deep neural networks. In this experiment, we test the generalizability under extreme conditions: whether the network is able to *directly* generalize to unseen input shapes even without a single adaptation process. We apply the model trained on Simple Shape Seen dataset and directly evaluate the performance on the Simple Shape Unseen dataset, which contains unseen shapes including line, triangle, and pentagon. Table 2 shows the quantitative performance of different approaches, and Figure 6 and 7, visualize the corresponding confusion matrices for shape classification and some qualitative examples, respectively.

We demonstrate that the graph-based approaches, including ours and the graph parser baseline, have the capability of generalization to unseen shapes. This is demonstrated by introducing the additional knowledge-based symbolic classifier. Even without a single exposure of a certain shape, one can describe the unseen

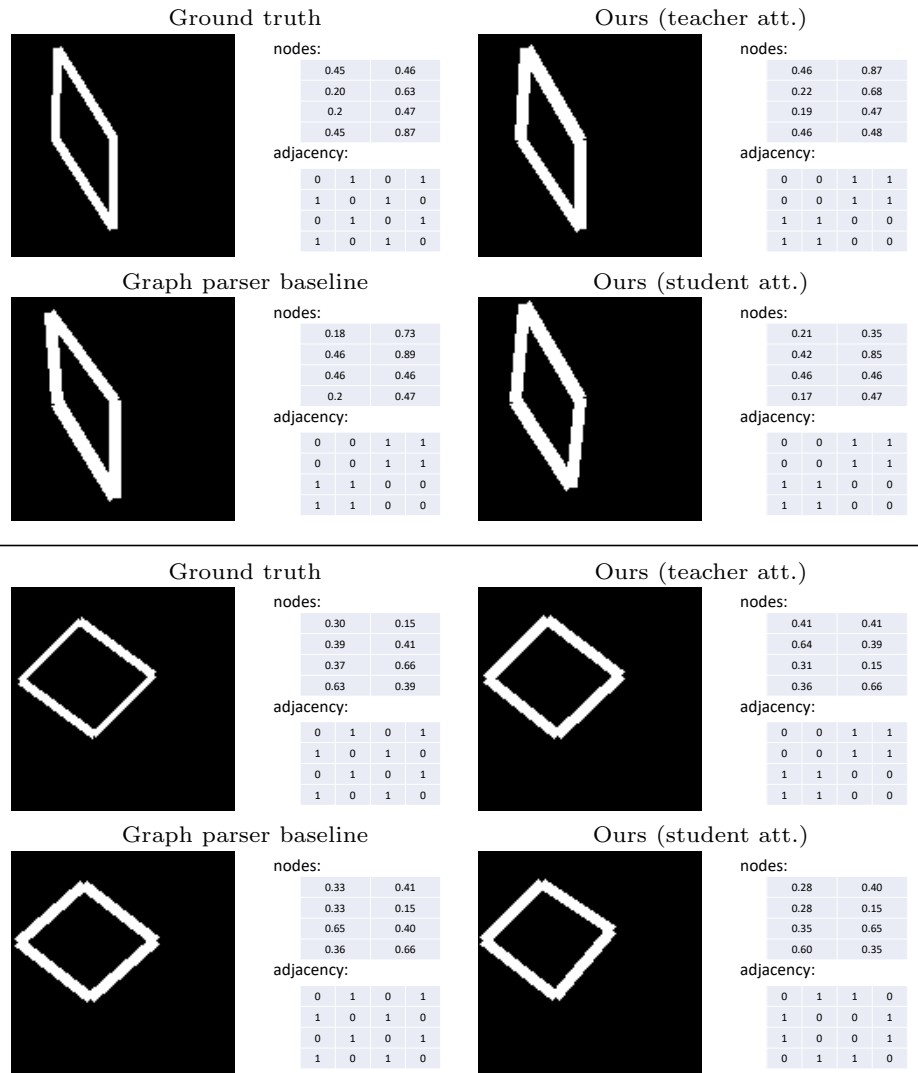

Figure 5: The qualitative results of graph-based approaches evaluated on *Simple Shape Seen* test set. We visualize two examples (separated by the horizontal line). The results of the CNN classifier baseline are not visualized since it directly predicts the shape labels.

shape's adjacency information in the symbolic system. Then, the task for the network is to generate the description of the scene, *i.e.*, the scene graph, such that it can be explained to and understood by the symbolic system. Together, the network and the knowledge-based symbolic classifier compose an explainable and generalizable neural-symbolic shape classifier.

Specifically, for all graph-based approaches, we use the same symbolic classifier that has the knowledge of three different shapes, thus the relative comparison is majorly between the different networks. Along with the shape classification accuracy we also present the local triplet matching performance, for a comprehensive evaluation. From Table 2 it can be noted that our network with the teacher attention module fails to predict correct graph representation for unseen shapes (54.5% F1-score and 1.3% classification accuracy). This is majorly because the teacher attention, although being able to learn the nodes by itself, has the number of nodes pre-defined and fixed. Thus naturally, it cannot generalize to other shapes with a varying number of nodes. In contrast, our approach with the student attention module is able to predict graphs with *arbitrary* number of nodes: no matter if the tested nodes are decreased (line, triangle) or increased

Table 2: Quantitative results of our approach and the supervised baseline evaluated on *Simple Shape Unseen* dataset. See Figure 6 for the detailed shape classification's confusion matrices of different approaches.

| Method | Sup. | Triplet matching | | | Classification |
|---|---|---|---|---|---|
| | | Precision | Recall | F1-score | Accuracy |
| Ours (tea.) | None | 48.4±5.2 | 62.8±3.7 | 54.5±3.6 | 1.3±1.1 |
| Ours (stu.) | None | **90.9**±4.7 | 83.0±5.3 | 86.5±2.8 | 67.5±3.2 |
| Baseline | Graph | 90.3±4.3 | **88.0**±6.9 | **88.8**±2.6 | **80.1**±3.8 |
| CNN classifier | Label | - | - | - | 0.0±0.0 |

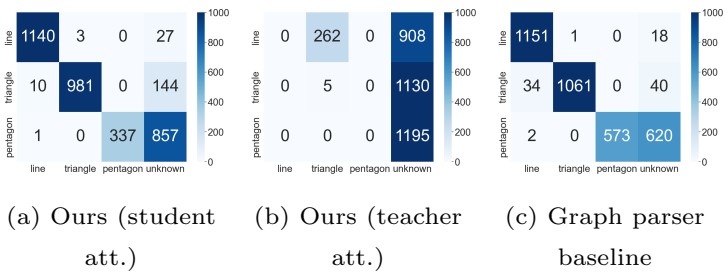

(a) Ours (student att.)    (b) Ours (teacher att.)    (c) Graph parser baseline

Figure 6: The confusion matrix of macro shape classification evaluated on *Simple Shape Unseen* dataset. See Table 2 for the overall accuracy of different approaches. The result of CNN classifier is not presented as it completely fails on the unseen dataset.

(pentagon). Interestingly, the graph parser baseline, although being able to generalize to unseen shapes and still having the best performance, fails to retain the large margin compared with our approach with the student attention module, when compared to the dataset containing seen shapes used in Section 5.1. The margin for F1-score and classification accuracy are reduced from 5.2% to 2.4%, and from 23.0% to 12.6%, respectively. Our approach even slightly outperforms the baseline in terms of precision. This further validates the generalizability of our approach when using the student attention module.

In contrast to the graph-based approaches (ours and the graph parser baseline), the CNN classifier baseline does not share the generalizability to unseen shapes. From Table 2, it can be observed that, the CNN classifier baseline exhibits 0% accuracy on unseen shapes, as it can only predict labels that are included in the training set.

**Re-training CNN classifier with unseen shapes** Given the failure of the CNN classifier on the unseen shapes, we perform one additional experiment, to fairly reflect the performance of the CNN classifier baseline and further showcase the generalizability of the proposed graph-based methodologies. We re-train the CNN classifier with a training set that contains an increasing portion of unseen shapes, to enable its functionality of predicting unseen shapes. After re-training, the accuracy is re-evaluated on the same Simple Shape Unseen dataset, and compared with two graph-based approaches (ours and the fully-supervised graph parser). The comparison is visualized in Figure 8, and as expected, as long as the CNN classifier is exposed to unseen shapes during training, it is able to provide satisfactory performances, and the more unseen shapes are included in the training set, the better the performance it can achieve. However, it can be noted that, when the number of unseen shapes is limited, *e.g.*, 0.5%, our graph-based approaches can still out-perform the CNN classifier, even though they are never trained with unseen shapes. It is worth to highlight that, in this setting that unseen shapes are included in the training set, the comparison between graph-based approaches and the CNN classifier is *never fair*. The fact that the graph-based approaches can still outperform the CNN classifier (when 0.5% of unseen shapes are included for re-training) further validates the generalizability of our approach.

Please note that we do not claim that the generalizability and explainability issues are fully solved by our graph auto-encoding method. Instead, we hope this research inspires more work in the direction of using

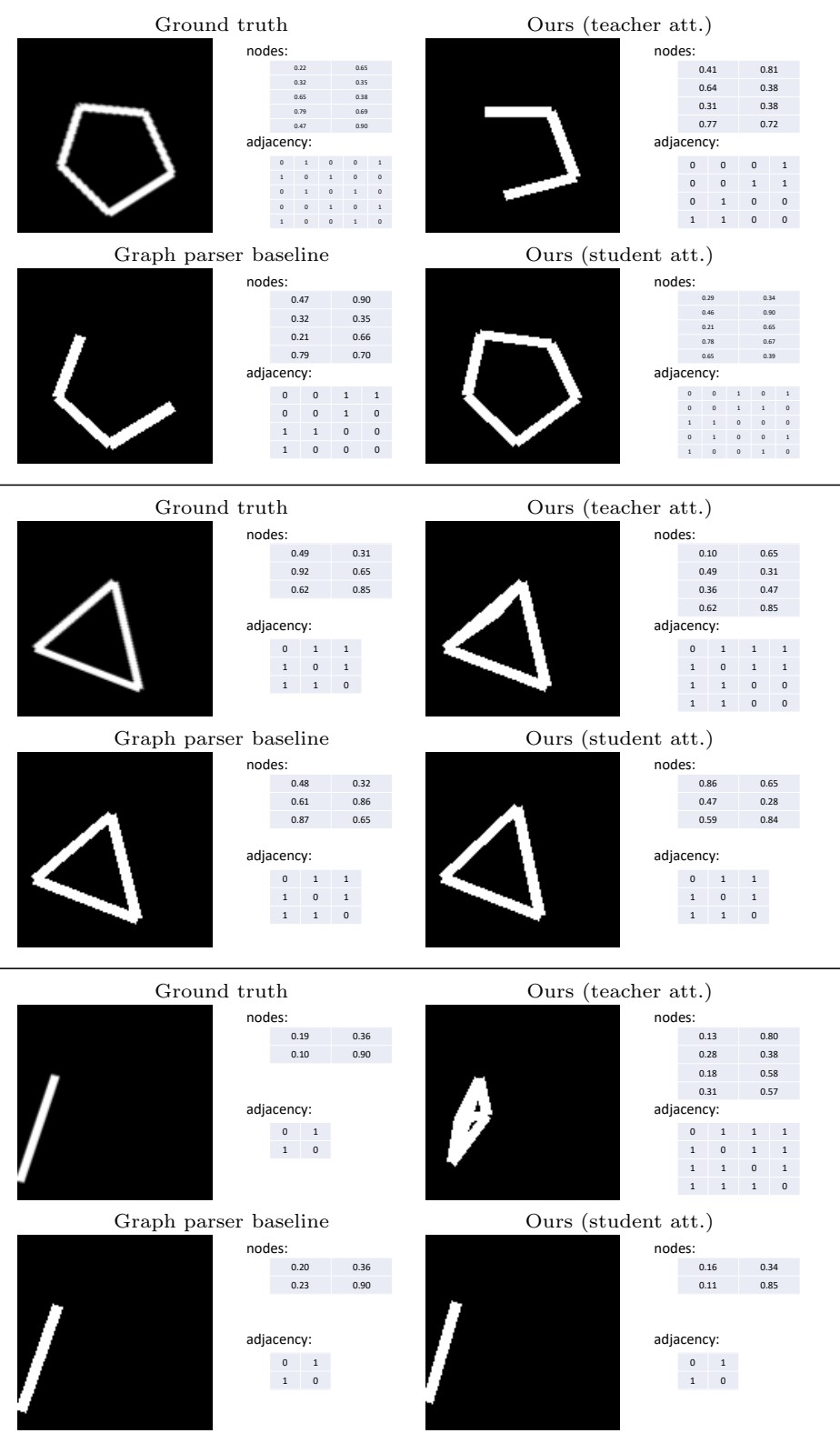

Figure 7: The qualitative results of graph-based approaches evaluated on *Simple Shape Unseen* dataset. We visualize three examples (separated by the horizontal lines). The results of the CNN classifier baseline are not visualized since it directly predicts the shape labels.

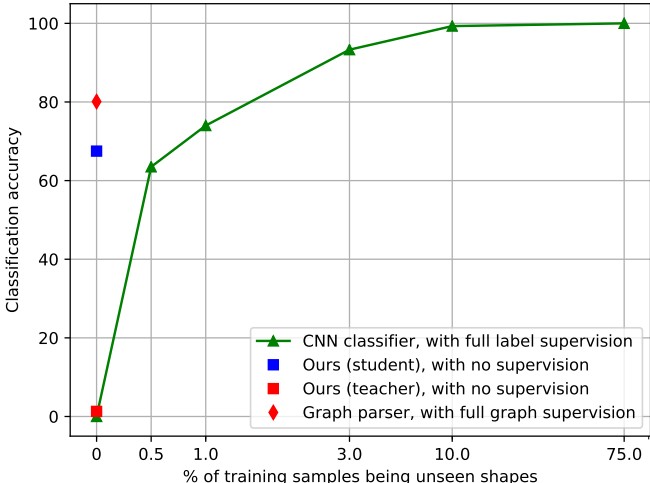

Figure 8: The performance of the fully-supervised CNN classifier tested on the *Simple Shape Unseen* dataset, when re-training it with unseen shapes exposed. The performances of our self-supervised auto-encoding approaches and the fully-supervised graph parser baseline are also visualized.

graphs as an interface between neural networks and symbolic systems, and thereby creates a path towards generalizable and explainable AI.

## 5.3 Ablation studies

### 5.3.1 Robustness to pre-defined number of nodes

In the previous main experiments, we set the pre-defined maximum number of nodes for the teacher attention module to 4, according to the property of the seen dataset (rectangle shapes). In this experiment, we study the performance change when the defined maximum number of nodes is redundant, which is typical for real tasks. Table 3 presents the quantitative results of different values for the maximum number of nodes. For the sake of simplicity, we only show triplet matching F1-score along with shape classification accuracy in this experiment, since the precision and recall are highly correlated to it.

From Table 3, it can be concluded that the redundancy of the maximum number of nodes will result in a performance drop: with the number of defined nodes increasing, the triplet matching F1-score and shape classification accuracy decrease accordingly, which holds for both teacher and student attention modules. This is mainly because, when the network has extra chances to reconstruct the image, the prediction tends to be undecidable. For example, to reconstruct a certain triplet in an image, if there are two extra triplets for the network to predict, the network cannot deterministically decide whether to generate one triplet with the confidence of connectivity being 1 or to generate three triplets with the confidence of connectivity being 1/3. Both choices will result in the same and correct coarse image, which inhibits the network to provide the single correct prediction.

When the node redundancy is limited, *e.g.*, 5 and 6, the performance can remain at an acceptable level, although the self-learned teacher attention module fails when the maximum number of nodes is very large, *e.g.*, 100% extra nodes or more. Furthermore, note that as long as the learning is successful, the student attention module can in theory handle an arbitrary number of nodes during inference, and the system is not restricted by the pre-defined number of nodes.

Table 3: Quantitative results of our approach evaluated on *Simple Shape Seen* test dataset, with different defined maximum number of nodes in the self-learned teacher attention module.

| # defined nodes | Triplet matching F1-score | | Shape classification Accuracy | |
|---|---|---|---|---|
| | tea. att. | stu. att. | tea. att. | stu. att. |
| 4 | **98.5**±0.7 | **93.3**±0.6 | **96.2**±0.6 | **66.7**±3.5 |
| 5 | 95.4±1.5 | 90.8±1.3 | 83.9±6.5 | 56.2±6.3 |
| 6 | 93.8±1.9 | 88.0±2.5 | 77.6±8.4 | 42.6±10.0 |

Table 4: Quantitative results of our approach evaluated on *Simple Shape Seen* test dataset, when trained with different losses.

| Reconstruction loss | Triplet matching F1-score | | Shape classification Accuracy | |
|---|---|---|---|---|
| | tea. att. | stu. att. | tea. att. | stu. att. |
| full | **98.5**±0.7 | **93.3**±0.6 | 96.2±0.6 | **66.7**±3.5 |
| w/o coarse | 98.0±0.6 | 92.5±1.9 | **96.6**±0.8 | 66.7±6.2 |
| w/o refine | 90.8±1.0 | 82.2±3.1 | 66.8±1.9 | 34.4±3.0 |

### 5.3.2 Loss ablations

We also perform the ablation study of different loss settings, to verify the design choice of the training loss. In addition to the loss setting introduced in Section 3.3, two variants of the loss are tested, with their performances listed in Table 4. Same as with the previous ablation study, we only show triplet F1-score and shape classification accuracy for the sake of simplicity.

Our default setting (full reconstruction loss) and the loss without coarse image reconstruction, see the first and second row in Table 4, exhibit similar performance in both metrics. However, one can observe that the standard deviation increases by a large margin when using the student attention module, if the coarse reconstruction loss is disabled: from 0.6 to 1.9 for F1-score, and from 2.5 to 6.2 for classification accuracy. This validates the necessity of the coarse image reconstruction, which stabilizes the performance when using the student attention module.

Instead of applying the main loss after the refinement sub-network, we also try to apply them only on the coarse reconstructed image that is directly provided by the differentiable CG module, without using the refinement network. From the third row in Table 4, it can be seen that the performance for every metric decreases by a large margin. This is mainly due to the domain gap between the training images and the online generated images from the differentiable CG module. This domain gap can also be observed in Figure 5. In this case, the supervision is not passed through the refinement network, thus the domain transformation is not performed. Since the (MS-)SSIM measures pixel-level similarity between two images, the domain gap will result in additional noise during the loss computation and thus inhibit the image reconstruction task, which should be domain-invariant and only focus on the graph structure itself. This experiment shows the importance of the refinement network and verifies that it can perform the domain adaption task during training.

## 6 Conclusion

By extending the auto-encoding paradigm, we provide a novel neural network design that can learn to extract visual elements from an image and encode them in a scene graph without the need for manually annotated ground truth during training. At its core, the scene graph's node and adjacency matrices are self-learned by properly designing the network architecture of the encoder, and aligning it with a decoder that is based on differentiable image drawing techniques. Furthermore, to demonstrate that the learned representations are

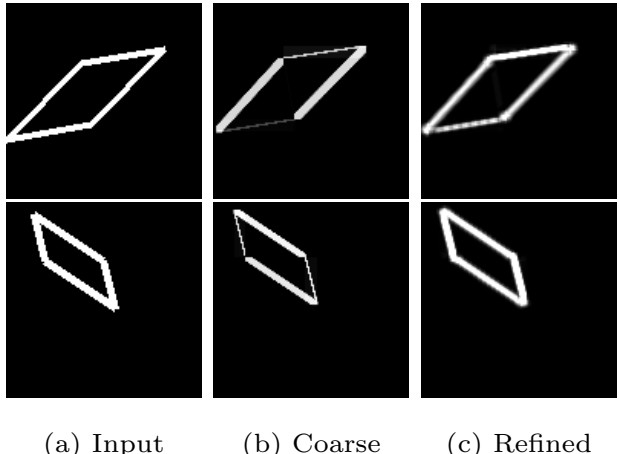

(a) Input          (b) Coarse          (c) Refined

Figure 9: Samples of the input images (a), images generated by the differentiable CG module (b), and images predicted by the refinement network (c). The domain gap can be easily observed between the input and coarse images, which is eliminated by the refinement network.

fully explainable, we apply them in an example task of knowledge-based shape classification. The knowledge-based classifier together with the learned encoder composes a rudimentary hybrid neural-symbolic system.

We acknowledge that key limitations exist for our current approach when scaling it to significantly more complex scene understanding tasks on natural images. For instance, the learning procedure of our unsupervised approach is less efficient than the supervised baseline, and the model design is more complicated and less straightforward. Furthermore, as a consequence of the aforementioned drawbacks, currently our model is limited to line drawings of simple basic shapes.

We evaluate our model with several experiments and ablation studies, and summarize the following potential further advantages of our approach. First, our hybrid neural-symbolic system is explainable, the graph predicted by the encoder is directly interpretable by humans and symbolic parsing systems. Second, since our network extracts visual elements from the image, it is naturally able to generalize to unseen shapes, without requiring any adaptation process. Together with the symbolic knowledge-based classifier, ideally one can classify any unseen shape. Third, our network is trained in a self-supervised manner and thus does not require any manual annotation, which greatly reduces the cost for training and deploying such systems. Forth, our annotation-free method comes with satisfactory performances when compared with two baselines.

In the future, we aim to extend the functionality of our model, especially in terms of the scene complicity of the input image and its corresponding graph representation. Although significant research is needed to reach this goal, we believe our work adds value to achieving more annotation-affordable, explainable, and generalizable neural networks, and hope it will spark more innovations in these lines of research.

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
