# OpenReview forum: "Image-graph-image auto-encoding: a conceptual study with symbolic shape classification"
_TMLR — Rejected by TMLR_

### Review · Reviewer_CUg7 · 2022-07-23

**Summary Of Contributions:**

# A. Summary
This paper proposes to learn scene graphs in an self-supervised (auto-encoding manner). The main idea is to have an auto-encoder architecture where the bottleneck layer forms a graph. They experiment on a toy dataset consisting of binary images of shapes and defer natural images to future work.


**Requested Changes:**

The empirical evidence is weak on a single synthetic dataset. The work could be strengthen by demonstrating the generalizability of the proposed approach by evaluating on more realistic datasets.

**Strengths And Weaknesses:**

# B. Strengths
Overall, the paper is easy to read and the motivation of the work is clear.

# C. Weaknesses

## C1. General

**C1a. Similar works [A,B] which also learn to construct scene-graphs are missing in related works.**

Both of these works learn the object concepts and the spatial relationship (which is a special form of scene-graph) for visual question answering. Albeit, they do have more supervision through the question and answer pairs. Personally, I think the setting in [A,B] is much more realistic and meaningful compared to the explored minimalist synthetic shape data.

- [A] Mao, Jiayuan, et al. "The Neuro-Symbolic Concept Learner: Interpreting Scenes, Words, and Sentences From Natural Supervision." International Conference on Learning Representations. 2019.

- [B] Shi, Jiaxin, Hanwang Zhang, and Juanzi Li. "Explainable and explicit visual reasoning over scene graphs." Proceedings of the IEEE/CVF Conference on Computer Vision and Pattern Recognition. 2019.

**C1b. The proposed approach is specialized for the designed data.**

It is difficult to see how it would generalize beyond the proposed dataset. For example, how would it work for MNIST. What would be the nodes, the edges? From the algorithm, I do not see any guarantee that the learned graph is the right level of abstraction for the down-stream task.

**C1c. It is unreasonable to assume that the objects and their relationship primitives are known (specific to the dataset) when designing the architecture.**

For example, the relation classification, it is simply checking where a line exists between the two corners. However, it is unclear how the proposed architecture could capture abstract classes when it is simply a binary classification. While the author claim

> it can be extended with semantic information with little extra effort

I generally disagree. In theory, "the model can be extended" by changing a binary classification to multi-class classification. But whether it would learn the right semantics is highly questionable. Also, the choice of spatial transformer network seems to be specific to this task (as the data is created with affine transformations). It is difficult to see the proposed architecture to work/inspire newer architectures on more complicated dataset.


## C2. Approach

**C2a. Implementation details are unclear/ deferred to an "Anonymous authors work".**

It is unclear why the authors are referring to a work by anonymous authors for which the reviewer cannot check the implementation details and no supplemental materials are provided. Hence, many details are missing from the work.

**C2b. Motivation of the teacher/student architecture is unclear.**

It seems like the main reason of using the teacher/student architecture is to

> in theory handle an arbitrary number of nodes during inference

However, for a typical dataset, the max maximum number of objects is likely not too big. Just as recent object detection architectures simply set the max number of proposals to a large number. Are there other motivations for using this teacher/student architecture?


## C3. Experiment

**C3a. The experiment is very limited in scope.**

The only dataset they evaluated on is a toy dataset consisting of synthetic dataset. The shapes are images of black and white images with a very clean background and no noise. It is unclear whether the results on this dataset are significant and its implications/contribution to future work.


## C4. Misc.
**C4a. Fig. 6 is awkwardly structured.**

---

> ### Author Response · Authors · 2022-08-17
> **Response to the Reviewer CUg7**
>
> We would like to thank the reviewer for the constructive comments and concerns. Our response is stated as follows.
>
> C1. General
> C1a. Similar works [A,B] which also learn to construct scene-graphs are missing in related works.
>
> We will add these works and improve the related work section in the revised version. We acknowledge that there are massive related works that tackle scene graphs and reasoning, and many of them work on realistic datasets. However, we start from a more challenging scenario: learning without any external supervision, which distinguish our work from others.
>
>
> C1b. The proposed approach is specialized for the designed data.
>
> We acknowledge that, in terms of applicability to challenging and realistic computer vision tasks and datasets, alternative and more mature methods are generally more capable and to be preferred over our novel graph auto-encoding method. However, we view our graph auto-encoding method as a fundamental contribution to the field of representation learning via auto-encoding, but whose real-world applicability is still to be investigated and determined in future work. Whether or not these properties are useful to applications of computer vision is subject to ongoing research.
> For the concern that “if learned graph is the right level of abstraction for the down-stream task”, we believe this should not be an issue. First, our original motivation is to demonstrate the capability of learning graphs in a self-supervised manner. To further demonstrate it, we propose this task. Therefore, we do not claim that the graph is most suitable for the proposed task or many other potential tasks. Second, the scene graph representation is already well received by the community. Theoretically, it has high-level abstraction and interpretability, which is arguably better suited for down-stream tasks which are often at higher abstraction level.
>
>
> C1c. It is unreasonable to assume that the objects and their relationship primitives are known (specific to the dataset) when designing the architecture.
>
> We agree that more limitations exist when learning relations in an unsupervised auto-encoding manner, compared to that in a fully-supervised manner.
> In fully-supervised learning, almost any relation can be learned as long as they are annotated in the ground truth and can be inferred based on visual cues. This holds for most of the existing work generating scene graphs from realistic datasets. In self-supervised auto-encoding, more limitations exist. The key is that the relation must be visible and can be identified via pixel-level variations. The relation proposed in this manuscript is one of the cases that meet this requirement. And we believe “it can be extended with semantic information with little extra effort, as long as the relation is visible can be identified via pixel-level variations.” These relations include visible semantic relations, such as proximity relation.
> We do agree with the reviewer’s concern and acknowledge the limitation of this learning paradigm. Some invisible relations that rely on high-level human reasoning cannot be learned, which requires human’s manual annotation anyway.
>
>
> C2. Approach
> C2a. Implementation details are unclear/ deferred to an "Anonymous authors work".
>
> We understand that in the manuscript it is not possible to cover all the implementation details. However, we will release the code and data once this manuscript is officially published, and therefore we hope the temporary unclearness should not be a problem.
>
>
> C2b. Motivation of the teacher/student architecture is unclear.
> In the paragraphs of Section 3.2.1 – Node attention, we have discussed the motivations of employing two attention modules. Briefly, the teacher module is able to learn to attend the nodes from the image reconstruction loss in a self-supervised manner, but it has a pre-defined maximum number of nodes to attend, which inhibits its generalization to images with arbitrary number of nodes. In contrast, the student module is able to generalize to images with arbitrary number of nodes, but cannot learn its weight from the image reconstruction loss, due to its detection-based design. Thus, the student learns from the prediction of the teacher during training, and is mainly used during inference.
> We cannot simply set the max number of proposals to a large number, since this will introduce performance drop if the redundancy of the defined nodes is significant (see the ablation study in Section 5.3.1). The self-supervised learning in the manuscript is much more challenging than the fully-supervised learning, as in most object detection frameworks. We will make this clearer in the manuscript.
>
>
> C3. Experiment
> C3a. The experiment is very limited in scope.
>
> This concern is very similar to C1b, and we have explained it in the response to C1b.
>
>
> C4. Misc.
> C4a. Fig. 6 is awkwardly structured.
>
> We will improve and reorganize the figure in the revised manuscript.

---

### Review · Reviewer_LjWc · 2022-08-04

**Summary Of Contributions:**

The paper studies effective ways to encode images into symbolic representations in a self-supervised fashion. The key idea is to design a differentiable auto-encoder that encodes numeric images into symbolic graphs, and that further decodes the symbolic graphs back into numeric images.

The study is restricted to the synthetic task consisting of simple shapes (represented as 2D images as the inputs). In this specific setting, the authors demonstrated that the learned symbolic representations are both explainable and generalizable through the task of classifying unseen shapes during inference.

**Requested Changes:**

The paper would be strengthened by either providing direct evidence that the proposed approach can generalize to more complex tasks, or providing convincing rationale that the task of 2D shapes is representative enough as a proxy to many real-world tasks.

**Strengths And Weaknesses:**

Strengths:

The paper is overall well written and easy to follow. The experiment setting is interesting -- despite limited scope, it offers a clean setting to build and study a working neural symbolic system. The setting could potentially be used as (part of) a synthetic benchmark for similar studies.

Weaknesses:

As the authors pointed out in the paper, the scope of the study is very restricted. The core assumption for this study to be useful in practice is that a similar system can potentially generalize to more complex tasks beyond 2D shapes such as natural images. However, this critical property (generalizability to other tasks) is not obvious yet from the paper:

* The proposed encoder-decoder is very specific to the underlying task (namely shapes). E.g., the decoupling of nodes (V) and edges (E), the differentiable coordinate system, and the spatial transformer network for edge rendering. This makes it unclear whether the approach or its extension is applicable to more complex settings (e.g., over a mixture of shapes and natural images), and even so, whether the new system with less task-specific engineering can retain comparbale results reported in the paper.
* While designing a symbolic representation is viable for simple 2D shapes (for which things like graph isomorphism tests are well-defined and inexpensive), designing a symbolic representation for other tasks may not be straightforward. E.g., nor the notion of nodes/edges or their mappings to the classes are defined for real images. Demonstrating the usefulness of the system over 2D shapes may have little to do with the hardest unsolved part of applying neural-symbolic systems in practice.

---

> ### Author Response · Authors · 2022-08-17
> **Response to the Reviewer LjWc**
>
> We would like to thank the reviewer for the constructive comments and concerns. Additionally, we thank that the reviewer finds experiment setting is interesting despite its currently limited scope. Our response is stated as follows.
>
>
> 1. The proposed encoder-decoder is very specific to the underlying task (namely shapes)…
>
> As indicated in the manuscript, we do acknowledge that the approach is limited in terms of task generalization. We view our graph auto-encoding method as a fundamental contribution to the field of representation learning via auto-encoding, but whose real-world applicability is still to be investigated and determined in future work. The goal of this manuscript is to present our novel fundamental method and demonstrate its properties. The experiments contained in the manuscript serve the purpose of demonstrating the soundness of our novel method and highlighting its conceptual and unique properties, i.e. being able to learn human-interpretable graph representations from image data without requiring manual annotations via auto-encoding.
>
>
> 2. While designing a symbolic representation is viable for simple 2D shapes…
>
> We agree that the current version of neural-symbolic system is relatively simple.
> Compared to the contemporary computer vision tasks and their neural-network-based solutions, symbolic systems exhibit diversity and often there is no canonical framework for most of the detailed tasks. From the perspective of general tasks, different symbolic systems need to be designed according to different use cases. It is not realistic to propose a specific symbolic system that solve the hardest unsolved part of applying systems in practice.
> Furthermore, it is not our intention to really demonstrate the usefulness of the neural-symbolic system over 2D shapes. The motivation is that the neural-symbolic system has the potential to overcome the interpretability and generalization issues, which are common in contemporary end-to-end neural networks. And we believe the graphs learned without any supervision could serve as an interface between the neural-network-based representation learning and various symbolic systems for downstream tasks. The proposed task is intended to be a good example, but not to be a generalizable solution.

---

> > ### Comment · Reviewer_LjWc · 2022-09-09
> > **Response**
> >
> > I'd like to thank the authors for their response. While experimental results in the simplified setting are encouraging, the path to generalize the same methodology to more realistic scenarios still remains non-obvious to me. My review remains unchanged.

---

### Review · Reviewer_tuf4 · 2022-08-04

**Summary Of Contributions:**

This paper proposes a model for parsing images of rectangles, triangles, pentagons, or lines, into vertices and edges, from which a graph can be built. The model is trained end-to-end as an autoencoder using a reconstruction loss. The model has multiple hand-designed mechanisms to deal with this task, such as a module which takes a softmax over feature maps to predict the location of vertices, and a procedure for distilling this module into another such module which also predicts the location of the vertices. Experiments are presented on a synthetic dataset of rectangles, triangles, pentagons, and lines. The authors evaluate the ability of their model to classify unseen shapes by extracting the graph and applying a hand-designed rule-based classifier for the previously unseen shapes.

**Requested Changes:**

-The CNN classifier baseline in the experiments serves no purpose whatsoever. In one instance, it is trained to classify a single class (for which there exist no negative examples), and in the other instance, it is then asked to classify 3 unseen classes which it is obviously incapable of doing. This experiment seems meant to demonstrate the fact that a 1-way classifier trained only on positive data cannot generalize to classes it has not been trained to recognize, but this fact is obvious–I would recommend the authors simply replace this portion of the experiments with the succinct statement “classifiers trained only on squares cannot recognize triangles, pentagons, or lines.”

**Strengths And Weaknesses:**

Strengths:

-The paper is fairly clearly presented.

Weaknesses:

This paper has two core weaknesses. First, the setting and task which the authors seek to solve is distant from any practical task of interest (such as, for example, scene graph extraction from actual data). Working on toy tasks is not in itself a problem, as toy tasks can often be used to derive valuable insights which aid in understanding actual tasks of interest. However, the methods in this paper are tightly coupled to the very specific toy task in play, and there is no reason to believe that the designed mechanisms are in any way reusable outside of this regime.

Take, for example, the student and teacher node attentions, which are trained to extract vertices. The equivalent of these for natural images would arguably be something like an object detector (indeed, several building blocks used in this model are borrowed from the object detection literature), and object detectors are common in the scene graph extraction literature, but if this is the case, then there is no reason to develop a less sophisticated (albeit still complex)  module whose only function is to extract vertices from simple polygons.

This general issue is further magnified by the authors’ focus on rule-based symbolic reasoning: there is no reason to believe that the authors’s ability to distill simple rules for classifying polygons is in any way generalizable or scalable to constructing rules for understanding the myriad interactions between real objects, and even in the toy task which the authors have set for themselves, the rule-based approach gets poor accuracy given the simplicity of the task. This weakness is further exacerbated by the authors’ frequent discussion of human-like reasoning and implicit suggestion that either a scene graph representation or a rules-based symbolic explanation of data is in some way more “human-like” or likely to lead to more “human-like reasoning.”  While automatic construction of interpretable rules is perhaps outside of the scope of this work, ignoring the fact that the presented approach has no way of being extended outside of the bounds of the toy task without major progress (relative to the presented hand-crafted rules) in a completely separate area weakens this work. The authors also ignore the long history of failure that rule-based approaches have been known to suffer in the area of machine learning (the GOFAI idea that we can distill human intelligence into a finite list of rules has seen diminishing popularity in the last several decades for a reason), and while this does not at all mean that the authors should not pursue this direction or that there aren't valid and valuable reasons to desire such approaches, it is important to be up front about our understanding of the inherent limitations of these approaches, rather than elevating them without such acknowledgement.

Even this on its own would perhaps not be too big of a problem, but the issue is once again compounded to an even greater degree by the fact that the paper seeks to establish itself as laying foundations for more complex work into scene graph extraction, but fails to engage with a huge quantity of relevant literature which already extracts scene graphs from actual complex scenes, as well as structured representations of various other flavors (see my references). There is a huge amount of work on extracting scene graphs from real complex images using substantially more sophisticated methods, and it is not sufficient for the authors to claim that because their method is perhaps the first which primarily is based on *autoencoder pretraining* that this is in some way exceptionally foundational, or that it gives them the right to ignore the majority of extant literature. A quick skim through the survey by Chang et. al (see my references) shows a rich range of approaches which scale to real data, and there’s even a Stanford CS236 class project from 2019 which employs an extremely similar approach (extracting a scene graph using an autoencoding approach), but which actually operates on real datasets.


The failure to properly contextualize this work is an issue that can be overcome by editing the related work section, but the content of the work itself is insufficiently connected to the current stream of scientific progress in this subfield. Introducing a new toy task which has not (by definition) been accepted by the community as a meaningful testbed means that the burden of proof that this task is relevant falls onto the authors; the authors must prove that insights gleaned from their testbed can be leveraged in more realistic scenarios which reflect the current state of progress in the field. Again, it is not necessary to work at large scales, to achieve SOTA results, or to demonstrate new capabilities, but work which is too distantly removed from practical progress needs to be valued and evaluated differently;  for instance, designing a toy task which can be rapidly explored and which is known to be a good proxy for improving understanding of more complex tasks, similar to work on c. elegans in other fields, would be tremendously valuable! But in this case the authors focus much more on the hand-crafted, extremely specific modules which only really seem to apply to the new synthetic task they have set for themselves. Overall, without a complete overhaul of the stated goals, intent, and content of the work, the simple fact is that the task the authors solve is too toy and their methods too hand-crafted for this to be a strong contribution or of interest to the community.


-This paper is much longer than it needs to be, and yet some critical parts of the method are still not actually explained (i.e. the only reference for how the node attention modules are actually trained is the code, not even an appendix). The content of this paper does not justify an 18-page body. The authors may wish to consider moving auxiliary content to an appendix, and focusing on producing a conference length paper of 8 or 9 pages.

---

> ### Author Response · Authors · 2022-08-17
> **Response to the Reviewer tuf4**
>
> We would like to thank the reviewer for the constructive comments and concerns. Our response is stated as follows.
>
>
> 1. The reviewer is generally concerned about the gap between the toy task proposed in the manuscript and the more challenging tasks that already exist, such as scene graph generation on natural images.
>
> We do acknowledge that, in terms of applicability to challenging and realistic computer vision tasks and datasets, alternative and more mature methods are more capable and to be preferred over our novel graph auto-encoding method. However, we view our graph auto-encoding method as a fundamental contribution to the field of representation learning via auto-encoding, but whose real-world applicability is still to be investigated and determined in future work. The experiments contained in the manuscript serve the purpose of demonstrating the soundness of our novel method and highlighting its conceptual and unique properties, i.e., being able to learn human-interpretable graph representations from image data without requiring manual annotations via auto-encoding. Whether or not these properties are useful to applications of computer vision is subject to ongoing research. As we view our method as a fundamental contribution, we believe TMLR is an excellent channel through which to share our findings with our academic peers.
>
>
> 2. The reviewer is also concerned about the limitations of rule-based symbolic reasoning.
>
> The presented rule-based symbolic reasoning is nothing but an illustration of how the self-learned graphs could be employed for high-level reasoning for various downstream tasks. We acknowledge this limitation and do not claim this is the right choice for the task of shape classification. The limitation of the symbolic reasoning is out of the scope of this manuscript. We do want to highlight that the graph is a well-recognized representation with significant advantages in terms of interpretability and generalization. It could be used in many other tasks, including various symbolic reasoning methods.
>
>
> 3. The reviewer is further concerned that the manuscript is not contextualized well enough with the related work.
>
> We do realize that there are massive previous research working on the scene graph generation task. From the perspective of task itself, it is fair to say that the proposed method performs the same image-to-graph transformation, but in a more restricted and limited settings (simple shapes). In contrast, most previous work can already do well on natural images, which reflects the weakness of our approach.
> However, as we stated in the response earlier and in the manuscript as well, it is not fair to directly compare our method to all previous work, since the supervision condition is significantly different, and our condition is way more challenging than the others. Therefore, we do not want to contextualize our work in the task of scene graph generation. Instead, we aim to showcase the graph can be learned via auto-encoding, and it is a high-level representation suitable for bridging the neural networks and symbolic systems.

---

> > ### Comment · Reviewer_tuf4 · 2022-09-05
> > **Response**
> >
> > Thanks to the authors for their response. Unfortunately, as might be expected from my review, I do not find the authors' arguments convincing in light of the content of the paper. In particular:
> >
> > > "we view our method as a fundamental contribution"
> > I would caution against making the claim that a contribution is in any way fundamental or foundational, especially in a field where there are already established foundations and fundamentals. If the authors wish to be able to claim that their work is a contribution to the fundamentals of the subtopic, they need to better characterize and contextualize the existing work to this end (again tying into my other concerns). The idea that the authors "do not want to contextualize [their] work in the task of scene graph generation" does not help their case in this respect even if their primary interest is actually in neurosymbolic systems, because the fact of the matter is that they have in fact chosen to produce something which is heavily tied in to this topic and not readily disentangled, not to mention that it is disingenuous to simply ignore a rich library of prior literature in a way which so conveniently elevates one's own work.
> >
> > My review remains unchanged, and I recommend rejection.

---

### Comment · Action_Editors · 2022-08-09
**Responses from authors to reviewer comments?**

I would be curious to hear how the authors respond to the strengths and weaknesses brought up by the reviewers. Some of the weaknesses highlighted by the reviewers are notable and I would like to understand in detail how you plan to address these comments. Please respond to each reviewer individually.

---

### Decision · Action_Editors · 2022-09-13

**Recommendation:** Reject

**Comment:**

This paper will not be accepted for submission to this journal. The central reason for this decision is that the claims made by the authors are not sufficiently convincing. All reviewers found the results unconvincing because the experimental design is too specialized and not well connected to the larger body of work in the field. The reviewers strongly encourage the authors to provide more connection for their toy example to the real world setting and identify opportunities for these experiments to generalize these results beyond the examples. I concur with this suggestion and encourage to rethink their experimental design in light of the large body of related work (cf. Chang et al, Stanford CS236 class).

Given the magnitude of changes suggested by the reviewers in terms of the experimental design and connection to the large body of literature in this arena, we will not consider a revised version of this manuscript for review. We do hope though that the authors use this feedback constructively to revisit their experimental design and identify new directions for connecting their work to larger body of literature in this arena.